# Value, transparency, and inclusion: A values-based study of patient involvement in musculoskeletal research

E. Bradshaw[1], K. Whale [2,3]*, A. Burston[2], V. Wylde[2,3], R. Gooberman-Hill[2,3]

1 University of Bristol and University Hospitals Bristol and Weston NHS Foundation Trust, Bristol, United Kingdom, 2 Musculoskeletal Research Unit, Translational Health Sciences, Bristol Medical School, University of Bristol, Bristol, United Kingdom, 3 National Institute for Health Research Bristol Biomedical Research Centre, University Hospitals Bristol and Weston NHS Foundation Trust and University of Bristol, Bristol, United Kingdom

* katie.whale@bristol.ac.uk

**Data Availability Statement:** Anonymised survey data is stored on University of Bristol Research Data Storage Facility and can be accessed via the University of Bristol Research Data Repository.

## Abstract

### Background

Patient and public involvement work (PPI) is essential to good research practice. Existing research indicates that PPI offers benefits to research design, conduct, communication, and implementation of findings. Understanding how PPI works and its value helps to provide information about best practice and highlight areas for further development. This study used a values-based approach to reporting PPI at a Research Unit focused on musculoskeletal conditions within a UK medical school.

### Methods

The study was conducted between October 2019 and January 2020 using Gradinger's value system framework as a theoretical basis. The framework comprises three value systems each containing five clusters. All PPI members and researchers who had attended PPI groups were invited to participate. Participants completed a structured questionnaire based on the value system framework; PPI members also provided further information through telephone interviews. Data were deductively analysed using a framework approach with data mapped onto value systems.

### Results

Twelve PPI members and 17 researchers took part. Views about PPI activity mapped onto all three value systems. PPI members felt empowered to provide their views, and that their opinions were valued by researchers. It was important to PPI members that they were able to 'give back' and to do something positive with their experiences. Researchers would have liked the groups to be more representative of the wider population, patients highlighted that groups could include more younger members. Researchers recognised the value of PPI, and the study highlighted areas where researchers members might benefit from further awareness.

Interview data is sensitive and contains potentially identifiable data and so cannot be shared. Access to survey data is restricted with data only available to bona fide researchers after a Data Access Agreement has been signed by an institutional signatory. Requests for access to the data may be made to data-bris@bristol.ac.uk.

**Funding:** RGH, AB, LB: National Institute for Health Research (NIHR) under its Programme Grants for Applied Research programme [RP-PG-0613-20001] https://www.nihr.ac.uk/ KW, RGH, VW: This study was also supported by the NIHR Biomedical Research Centre at University Hospitals Bristol and Weston NHS Foundation Trust and the University of Bristol. The views expressed are those of the authors and not necessarily those of the NIHR or the Department of Health and Social Care. https://www.nihr.ac.uk/ https://www.bristolbrc.nihr.ac.uk/ The work of PEP-R has been variously funded in grants from the National Institute for Health Research and Versus Arthritis. https://www.nihr.ac.uk/ https://www.versusarthritis.org/ The funders had no role in study design, data collection and analysis, decision to publish, or preparation of manuscript.

**Competing interests:** The authors have declared that no competing interests exist.

**Abbreviations:** PPI, patient and public involvement.

## Conclusions

Three areas for development were identified: (i) facilitating researcher engagement in training about the value and importance of PPI in research; (ii) support for researchers to reflect on the role that PPI plays in transparency of healthcare research; (iii) work to further explore and address aspects of diversity and inclusion in PPI.

## Introduction

Patient and public involvement (PPI) is considered to be a key part of good research practice [1, 2]. In health research it is increasingly recommended that patient representatives are involved at all stages, including identification of research priorities, study design, study management and oversight, communication of results, and planning next steps [3]. PPI helps to ensure that patients' voices are reflected and addressed in the design and delivery of health research and helps the outputs to be relevant and meaningful to them [4].

Recognition of the importance and value of PPI in research has grown substantially in the last decade. Funding bodies such as the UK National Institute for Health Research (NIHR) include a mandatory section on PPI activities as part of the grant application process in applied research. The UK Department of Health and Social Care supports PPI through funding INVOLVE [1], a national advisory group that supports active public involvement in the NHS, public health, and social care research; INVOLVE provides guidance on best practice. In 2020 the NIHR has launched a new Centre for Engagement and Dissemination which brings together activities in PPI, engagement, and participation with its strengths in research dissemination [5].

Literature on the impact of PPI demonstrates benefit to enrolment and retention in clinical trials [6], study design [7, 8], and diffusion of health-care innovation [9]. However, some have suggested it is challenging to objectively evidence the full scope of PPI impact because the wide range of activities and levels of patient involvement used in research can make it hard to evidence [10, 11]. In addition, early guidance about evaluation of PPI has focussed on practical elements of the process and lessons for good practice [8, 12], with limited evaluation of theoretical components [13, 14].

Discussions of PPI have often focused on reasons for including PPI in research. Recent literature has highlighted three themes: morality, transparency, and efficiency. First, patients have a right to have a voice in the research work that is conducted either about them or about treatment of their health, and that researchers have a moral obligation to redress any power imbalance in research and healthcare decision making through PPI [15–17]. Second, that PPI takes research outside of academia and into the real world [18, 19], which enables accountability and increases transparency [20]. Third, commentators write that inclusion of patients and members of the public in research planning and conduct makes it more efficient, more valuable, and means that findings are more likely to be successfully implemented in real-world settings [21–23].

In assessing the impact of PPI, it has been suggested that understanding both the theoretical aspects of PPI and the practical application of patient involvement work provides a fuller picture of how involvement functions [24]. Using a theoretical framework to underpin reporting of PPI is one way to achieve this, enabling the scope to review the value of PPI and to identify any areas for further development. A recent systematic review by Greenhalgh and colleagues identified the existence of a 'plethora' of frameworks, tools, and guidelines that exist to

support, evaluate, or report PPI [25]. Based on the review, the authors developed a taxonomy of five main categories of framework used in evaluation: power-focused, priority-setting, study-focused, report-focused, and partnership-focused. Importantly Greenhalgh and colleagues recommended that researchers consider which framework is the best fit for their purposes, and make decisions accordingly, ideally in collaboration with stakeholders.

## PPI and the Musculoskeletal Research Unit

To deliver PPI within the Musculoskeletal Research Unit at the University of Bristol, UK, the 'Patient Experience Partnership in Research' (PEP-R) group was established in 2010. PEP-R comprises patients with experience of musculoskeletal conditions and treatment. Since 2010, PEP-R has been expanded to reflect growth in research at the Unit, and at the time of writing three groups run alongside one another to provide specialised input into research studies that relate best to their own experience. Groups currently comprise: a group addressing a variety of musculoskeletal conditions and care (PEP-R), and two specialised groups addressing pain (PEP-R STAR) and infection after joint replacement (PEP-R INFORM).

A focused evaluation of the PEP-R group looking at the impact of patient involvement was conducted between 2011–2012, after the first/original PEP-R group had met ten times [26]. The evaluation was particularly focused on process-related aspects of the PEP-R group. On the basis of that evaluation we worked to improve research projects' feedback to the group on how their input had helped to shaped studies as well as enabling studies to engage as early as possible with the group. We wanted to use a theoretical framework to develop our understanding of PPI once the PEP-R activities were well embedded into the Research Unit and the two specialised groups had been running for several years, we therefore carried out a study of the three PEP-R groups at the Unit in 2019–2020. Following recommendations from Greenhalgh and colleagues we reviewed the existing evidence-based approaches to reporting PPI. We identified Gradinger's framework as appropriate and fit for purpose because of the focus on value systems that would enable us to explore meaning as well as process. Gradinger's work identifies value systems in public involvement and comprises three 'broad value systems': i) normative perspectives concerning ethical and political aspects of public involvement, ii) substantive perspectives relating to the consequences of public involvement, and iii) process values concerning conduct within public involvement for example, respect and trust and honesty. Each of these values are further split into five sub-clusters, which can be seen on Table 1. When used to understand involvement with patients and researchers, the framework gives a structured approach to describe and compare the values that occur in processes and activities of PPI in the PEP-R groups.

## Materials and methods

The study comprised questionnaires with researchers and clinical academics, and questionnaires and interviews with members of the PEP-R groups. Data collection took place from September 2019—January 2020. Ethical approval was granted by the University of Bristol Faculty of Health Sciences Research Ethics Committee in 2019 (reference: 91543, approval date: 03/09/2019) and all participants provided informed consent to take part. The design of the study was discussed with PEP-R including identification of their preferred methods for data collection. Group members suggested that additional telephone calls would enable them to expand on their questionnaire responses. Therefore, telephone calls were included in data collection after they had completed questionnaires. Initial results were discussed with three members of the PEP-R group and all three felt that the results accurately reflected their views. Study results were presented at a PEP-R meeting as a means of sense-checking (respondent validation) the

Table 1. Mapping to Gradinger's value system.

| Value system | Value cluster | Data |
|---|---|---|
| Normative | Empowerment | PEP-R questionnaires and interviews |
| | Rights | PEP-R questionnaires and interviews |
| | | Researcher questionnaires |
| | Change/Action | PEP-R questionnaires and interviews |
| | | Researcher questionnaires |
| | Accountability/transparency | No data |
| | Ethical Values | PEP-R questionnaires and interviews |
| | | Researcher questionnaires |
| Substantive values | Effectiveness | PEP-R questionnaires and interviews |
| | | Researcher questionnaires |
| | Quality/relevance | No data |
| | Validity/reliability | No data |
| | Representativeness/objectivity/generalisability | PEP-R questionnaires and interviews |
| | | Researcher questionnaires |
| | Evidence base | PEP-R questionnaires and interviews |
| | | Researcher questionnaires |
| Process values | Partnership/equality | PEP-R questionnaires and interviews |
| | | Researcher questionnaires |
| | Respect/trust | PEP-R questionnaires and interviews |
| | Openness/honesty | No data |
| | Independence | No data |
| | Clarity | PEP-R questionnaires and interview |
| | | Researcher questionnaires |

work. The group members thought that the results were insightful and of importance. We also sense-checked results with researchers at a Research Unit presentation, they thought that the results were a reflection of their experiences.

## Data collection

To design the questionnaire we used Gradinger's framework [27] with items tailored to the context in which the PEP-R groups were working so that questions made sense to them. Members of PEP-R were also asked if they would like to take part in a short telephone interview after completing the questionnaire, with these interviews designed to give participants the chance to speak in more detail about their views and experiences. Questionnaire responses for PEP-R members were recorded on paper and via email and returned to the research team. The interview topic guide is included as a S1 Table. Interviews were audio-recorded and transcribed verbatim. For researchers, completion was through Online Surveys, a portal that allows completion and collation of results online. Questionnaire responses from both groups and interview transcripts were uploaded to NVivo for analysis.

## Participants

The study included two groups of participants: (1) staff members comprising researchers and clinical, research active, academics with experiences of attending PEP-R groups at the Musculoskeletal Research Unit, (referred to as 'researchers') and (2) members of the PEP-R groups. Researchers and clinical academics who had attended PEP-R meetings during the previous three years were invited to complete the questionnaire. Responses were anonymous so that

participants felt able to provide candid feedback. Overall, 24 researchers were invited and 17 completed the questionnaire. All 17 PEP-R members were invited to complete the questionnaire. Twelve members returned questionnaires, 10 on paper and 2 electronically. All twelve who returned the questionnaire also took part in telephone interviews. PEP-R members were aged between 49 and 79 years old, and all had been members of the group for between 4–9 years. Due to the small sample size a detailed breakdown of demographics is not provided in order to maintain anonymity. We do not provide demographic details of the researchers because we asked them to provide information anonymously, this was because our consultation with researchers during the design of the study indicated that researchers would feel more open and candid to share their views if anonymous, and collection of even straightforward characteristics had the potential to identify them.

## Analysis

Questionnaire data and interview transcripts were imported into the qualitative software package NVivo 10. Data were deductively coded against constructs from Gradinger's Framework. Two transcripts were double coded and the data team worked together to compare data and enhance triangulation. We used framework approach as a means to structure the process in which we assigned code labels to portions of the data and then worked as a team to make sense of the material as a whole [28]. In brief, data were coded based on the three overarching value systems, each containing five clusters as shown on Table 1. Two researchers coded the data (AB & EB). Coding was discussed and refined within the research team to maximise rigour [29].

## Results

Data coded onto all three value systems, encompassing 11/15 value clusters. An overview of the mapping is shown in Table 1.

## Normative values: Concerning ethical and political aspects of public involvement

**Empowerment.**  Empowerment values relate to transfer of control, self-help, and the right to representation and accountability. Views of PEP-R group members primarily focused on representation and whether they felt empowered to share their views. Group members reported feeling confident to share their views in meetings.

> "*Basically, it's an open discussion, where I feel you can give your views, my views, whether it's ridiculous or not. It's quite open basically.*" (Patient 5)

Some members acknowledged that it took time for their confidence to develop and feel secure in expressing their views.

> "*I feel more comfortable now to make suggestions, but I was terribly nervous at first!*" (Patient 4)

Other members, particularly those who had been impacted by their health, thought that involvement in the group helped them to regain their confidence by engaging with others in similar situations. They described feeling empowered to contribute based on their own experiences.

*"It gave me my confidence back again . . . when you're at home after you have the operation it's a horrible feeling of being on your own. You don't know what's going to happen next, if anything. . .. So yes, it's been nice to see other people, how they managed."* (Patient 4)

**Rights.**    Rights values relate to the intrinsic value of PPI and the right to have a say. In addition to feeling able to express their views, PEP-R members felt that their voices were valued and that their contributions to research projects had an impact.

"*researchers come back and talk to us again and . . . tell us what's happened after talking to us the previous time, and how they've taken on board what we've said, and they've changed . . . things. It does seem to me like we are listened to."* (Patient 3)

Researchers also recognised the value of the group, but this was balanced by comments about the requirement to engage in PPI activity due to their job role or the expectations of research funders.

"It was a requirement of my role but also an education and a positive pressure."

(Researcher 7)

*"[It's a] r*equirement of the funding body." (Researcher 15)

Researchers discussed how to listen to patients' views. Some researchers felt that patients had more of an empowered voice in specific PEP-R groups than they did as patient representatives within study steering groups. However, their own preferences did not always map onto the expectations of funders.

"PEP-R tend to prefer taking research to the patient rather than joining as co-applicants on the [trial management groups], but we get a lot of pressure from [funders] to include as co-applicants." (Researcher 9)

"I think sometimes they can feel overwhelmed and disempowered by the conversations that occur between academics." (Researcher 8 about steering groups)

**Change/Action.**    Change/action values relate to generation and translation of knowledge into action to bring about change. PEP-R members showed high awareness of the impact the group had on research and the changes that were made. In particular, PEP-RI members made frequent reference to changes to documentation and plain English summaries.

*"The change that we've seen is the way some of the publications have gone out. Where before it was quite medically worded, now as we said put it in plain English so patients can understand. I think that's one of the biggest changes I've seen."* (Patient 11)

*"Well . . . there was a [patient information] leaflet. When we first look at them, we think, "No, that's not.. . . It doesn't make much sense," . . . there are also things that get added that we feel were important that haven't been included [in the first instance]."* (Patient 10)

PEP-R members were glad to receive feedback from researchers on the changes that had been made based on their input. One member reflected on longer-term dissemination work and how research participants wish to be kept informed, even after projects end.

*"It's quite important though, if you've got patients [who have] spent time being part of the research, it is important that those people do get some proper feedback on the outcome of the research. If that's been promised I think it's important to make sure it happens."* (Patient 8)

**Ethical values.** Ethical values relate to awareness of the need to protect PPI members from potential harms. Two aspects were discussed, the role of the group in providing emotional support and minimising potential distress and ensuring that researchers are keeping to good practice when conducting research.

Patient feedback focused on the social support they received from being in the group.

*"We've all had different experiences, we've all gone through different things, but we all understand where we're all coming from. So I think it is the support of the others in the group."* (Patient 11)

Some PEP-R members also talked about how being involved in the group had helped them process difficult experiences they had been through with their health conditions and create a positive experience.

*"I've been through it and it is the most horrible thing that you can go through. If something good and something positive comes out of it like one way is better than the other that is just brilliant . . . it's more than worthwhile."* (Patient 11)

Researchers focused on methodology and rigour. Researchers reported that involving PEP-R groups helped them to review their methods and make their research and patient-facing documentation better.

*"The feedback I gained in the groups directly impacted on both of my projects. I made changes to planned methodology, such as the length of data collection and formats of data collection. I made changes to the plain English summary and patient information booklet."* (Researcher 11)

## Substantive value systems: Focused on concerns about the consequences of PPI in research

**Effectiveness.** Effectiveness values relate to PPI having an effect on research and implementation. Discussion with PEP-R members focused on awareness of the wider implications of their input, beyond specific research projects, and how their input influenced dissemination activities. Group members felt that their own experiences had value and impact in terms of the feedback they could provide to researchers and health professionals.

*"Because we are the general public, we know what we went through, what would help us, what would have improved our wellbeing. We can say to the researchers and the surgeons, "If you'd done x, y and z it would have helped."* (Patient 11)

However, the real-world impact of their input and feedback was less clear to them.

*"Whether it results, then, in other people kind of changing what they would, like doctors in other hospitals; whether they would change what they do because of it? That's the bit that was probably the grey area."* (Patient 9)

This was also echoed in discussions about impact on the wider scientific impact.

*"Really, I'm not sure about the scientific impact that I would have."* (Patient 3)

Researchers also responded with comment about the effectiveness of PEP-R, through reflection on its impact on their work and the changes that they had made, this particularly centred on the details of the research design and associated data collection methods.

*"I made changes to planned methodology, such as the length of data collection and formats of data collection."* (Researcher 10)

**Representativeness/Objectivity/Generalisability.** This value relates to creating representative, objective, and generalisable knowledge through PPI. Patient group members and researchers were asked whether they felt the group was representative of the general population and whether it was diverse.

Views on the diversity of the group varied substantially between researchers and patient members. Diversity meant different things to each group. Researchers focused on the group demographics and whether they reflected the diversity of the patient population. They also talked about the level of experience that the group had, particularly in terms of benefits and drawbacks. Researchers appreciated the expertise of the group in terms of understanding research processes and what was needed from them at different stages of the research process. However, researchers also highlighted that this level of experience meant that the group had an elevated level of understanding compared to the general patient population, for example in relation to research language and methodology.

*"All members are now very expert in research. This is helpful but also not a true reflection of patient understanding in many cases."* (Researcher 11)

One researcher discussed the inclusion of patients in steering group meetings and the impact of having one expert patient.

*"The drawback is that there's a risk the patient is treated as a research expert, or that one person's experience becomes the benchmark by which important decisions are made. I'm not saying that's happened, it's just a risk."* (Researcher 10)

Patients' views about diversity focused on individual experience. Most considered the group to be diverse due to the range of experience and individual differences and saw value in these.

*"We've had just about everything going, I think, and ongoing infections, so, yes, we're covered in just about every spectrum I think."* (Patient 10)

Some patients highlighted that the age range of the group did not include many younger people. They wondered whether the working commitments of younger people might inhibit their ability to participate.

*"I'd like to see younger members of the group . . . that's always been my hope that we got a few more youngsters, but it's asking a lot. I mean, unless you're in a job where they can also see the value of it and will give you the time off, it's not easy."*

(Patient 6)

Others also recognised the age range of the group but thought that this was appropriate given that the focus of the group was health conditions and procedures that affected older people more.

*"We are a fairly diverse group . . . the sort of research that we're involved in, hip and knee replacements and all that sort of thing tends to be something that happens, generally, to older people. So, the group is going to tend to be older. [. . .] that's probably because younger people tend not to have hip and knee replacements unless they're very unlucky." (Patient 8)*

## Process value systems: Focussed on concerns about the conduct of PPI in research

**Partnership/Equality.** Partnership/equality value relates to the sharing of power of decisions in equal, reciprocal, and collaborative PPI processes. Patient feedback focused on the internal dynamics of the group and the balance between members, issues around shared decision making with the researchers, and views on group responsibility.

Group members reported high levels of equality and feeling supported and respected in sharing their views.

*"Especially between the members, you know, I know that anything I say is respected."* (Patient 9)

*"(the researchers) are all very friendly, and it's easy to say what you think. Nobody makes you feel that you're being foolish because of your lack of knowledge about the subject." (Patient 8)*

Many of the patients had been part of the group for a long time. Group members and researchers thought that this stability was an important consideration with pros and cons. This included the comfort that came from membership of a stable group, that was balanced by the potential challenge of integrating new members into an established group.

*"I can see an argument that says you need some sort of turnover of membership in the group. There are pros and cons of course, and I . . . feel increasingly comfortable in expressing an opinion with the people you know. Whereas, if you don't, then, perhaps, a bit more hesitant. On the other hand, if you've got a stable group like this and you introduce one new member that new member is bound to feel a little bit of an outsider, for a while at any rate."* (Patient 8)

When talking about achievement of consensus, patients talked about how opinions differed and how views were taken into account.

*"Not a big conflict, no. There were differences of opinion, but no."* (Patient 9)

*"Quite often the researchers will just take what we say, and I think then they change whatever we suggest. It's rare that anybody disagrees with us or says, 'No, we can't do that,' or whichever." (Patient 10)*

However, some group members did talk about feeling a sense of hierarchy in the group and said that this meant that they felt less able to contribute.

*"I do feel I'm a bit less intelligent than some of them that are there. You know, there are teachers and all sorts there, whereas I only went to a secondary school and didn't get any exams or anything, so sometimes I do feel as though I don't contribute enough. . ." (*Patient *7)*

However, feelings of difference between group members and the researchers was thought to be expected because of different degrees of research methodology and health knowledge (being 'better at it') between researchers and group members. For instance, one member spoke about this in relation to researchers and clinicians.

*"Well, I suppose we aren't equal because they're much better at it. But I don't feel minor to them if you know what I mean."* (Patient 7)

Feelings of differing levels of responsibility were also evident in decision-making. Group members did not think that it was their role to make decisions but rather to provide researchers with their views and opinions.

*"Future patients are in their hands, aren't they, so it's their (*researchers'*) decision."* (Patient 1)

*"I think, as a group, we're involved in the decision-making only because we are putting forward our thoughts. Obviously it's up to the researchers to use that in any further research, or, perhaps, make them think about, "Oh, I hadn't thought about that." As I said, I don't feel that we are making decisions as a group jointly at all. We're not taking a vote on anything. We're just free to express our own personal opinions."* (Patient 3)

**Respect/Trust.**   Respect/trust values relate to respecting diversity, values, skills, knowledge, and experience in mutually beneficial PPI processes. Researchers and group members felt that, without exception, there was respect and trust within the group.

*"When we are all in the room we all respect each other, everybody is there to listen to each other, we all have our own opinions and we all respect each other's opinions and that is where the trust comes in. The trust of it is we know we can say whatever we're feeling, and nobody will judge."* (Patient 11).

Group members reported feeling that all views and opinions were respected both within the group and by researchers who took part in the group.

*"I know that anything I say is respected . . . my comments are taken on board and, with anybody, with the other people in the group or the researchers. I don't ever feel any sort of disrespect at all."* (Patient 10)

*"They [researchers] do treat us as equals and always show that they value our opinion."* (Patient 7)

**Clarity.**   Clarity values relate to clarity of the purpose, processes, communication and definition of PPI. Group members and researchers demonstrated great clarity in the purpose of PPI. Researchers highlighted the value of involving patients' perspectives.

*"The group tends to have a different perspective to those of the researchers, and what a clinician considers important ends up being different to the patients."*

(Researcher 15)

The group valued clear communication about how PPI input was being used. As within the change/action value, feedback on any changes that had been made was key.

*"people come with their new ideas and their research, and I think they really do listen. And they come back again and say, 'Right, we've taken stock of this, and now we've done it this way.' you do feel really quite chuffed I suppose. . . at the end; you think, 'Oh right,' you know, 'they have changed that,' yes."* (Patient 6)

Group members talked about the importance of feedback about their input This focused on receiving information about how their views had influenced the research and that this maintained group member's interest and engagement.

## Discussion

This study provides insight into PPI work at a Research Unit mapped against Gradinger's value system framework [27]. Results show that PPI work mapped onto all three value systems, including eleven out of fifteen clusters.

PEP-R members felt empowered to share their own experiences and views on research projects. The supportive nature of the group and rapport between the members helped them gain confidence over time and feel valued by the researchers. These findings resonate with the morality-based rationale for PPI in research, particularly the right to 'have a say' [15–17]. However, this was implicit rather than explicit as although members of PEP-R said that they felt confident to voice their opinions in PPI activities they did not articulate strong feelings about their 'right' do so. Instead, they spoke about the importance of feeling valued and appreciated, and that they were doing something positive with their own experiences.

Researchers' views were mixed and predominantly aligned with the efficiency values of PPI work [21–23]. Researchers recognised the value of PPI, especially in reviewing patient documents and addressing methodological challenges. However, some felt that the main reason for their engagement in PPI activities was because of the requirements of research funders in the grant application process. Similar differences between patient and researcher motivations have been found in other studies. For instance, in a recent evaluation of PPI in a randomised trial of urinary tract treatments patient motivations focused on making a positive difference and supporting future patients, whereas researcher motivations were focused on improving research methods and enhancing their career opportunities [30]. In-depth qualitative work on health researchers' attitudes towards PPI has further highlighted differences in approach and value given to PPI, ranging from cynicism to ambivalence to positivity [31, 32]. Boyland and colleagues suggest that disciplinary background may be a factor and that hierarchy-based disciplines may place less value on patient perspectives, or even feel that their own expertise was undermined [31]. In addition, their work highlighted that responsibility for PPI responsibility often fell to more junior research staff who may not have the power either to embed PPI within research or to ensure that feedback is acted upon.

Our study highlighted group members and researchers had different views about diversity of the group. PEP-R's views on diversity centred on the value of individual health experiences and age range of group members, particularly the relative absence of younger people as current group members are between 49 and 79 years old. However, this reflects the typical age for joint replacement in the UK, with National Joint Registry 2020 showing that the median age for patients undergoing hip replacement is 69 years (interquartile range 61–76 years) and for knee replacement is 70 years (interquartile range 63–76 years) [33].

Researchers spoke about sociodemographic diversity and they suggested that they would value inclusion of members with a wider range of socio-economic and cultural backgrounds, and ethnicities. This was framed both in terms of whether the PPI group was representative of the wider population, but also the need to include underserved groups in research projects and PPI work. This is of particular importance in light of concerns that lower socioeconomic status and health inequality is linked with poorer health outcomes [34–36], including musculoskeletal conditions [37, 38]. Our work highlights the difference between a need to include diverse voices in PPI work and inclusion in research. We see these concepts as different but related. Inclusive PPI enables underserved voices to be amplified and heard in the processes of research design to ensure that the research meets everyones' needs. In research studies, representation ensures that the study includes members of the most appropriate population. This means that underrepresented groups are included and that the research findings are 'generalisable' from the sample to the population. Although the concept of generalisability in qualitative research is debated, and some prefer to refer to 'transferability' [39], all research seeks to generate knowledge of broader relevance and applicability. PPI on the other hand is a process and activity centred on design of research to meet this end goal, and ensuring diversity in this design is a way to improve research. Prioritisation and community-engagement activities that were precursors of PPI are good examples of approaches that seek to ensure inclusion and diversity. For instance, Citizens' Juries work by promoting active citizenship through processes of 'deliberate democracy' [40] and can bring together diverse citizens to discuss issues of public concern, such as health policy decisions [41, 42]. This approach has been used in priority setting for health research [43] and can enable engagement from members of the public [44]. In many ways, PPI takes a similar approach, seeking to include, empower and amplify in order to bring together members of professional and public communities, ideally for the public good.

Providing information on diversity in PPI groups presents ethical challenges around privacy and confidentially, and raises issues relating to disclosure. As the PEP-R group is small, provision of diversity information to external audiences may not be appropriate for reasons of confidentiality. Similarly, researchers who engaged with the PEP-R groups were not provided with access to personal and confidential information about group members' protected characteristics or circumstances, unless group members choose to disclose such information directly to them. As such, researchers may be basing their well-intentioned feedback about diversity without full knowledge of the group members' characteristics or circumstances. We consider it to be crucial that inclusion of underserved groups is fostered and prioritised in all aspects of research. We found it interesting that views of what diversity means and the diversity present within PEP-R varied between researchers and group members and suggest that diversity and inclusion in PPI could be a topic for further research.

The importance of diversity in PPI groups has recently been stressed. A 2020 NIHR report has re-emphasised the need for greater equality and diversity in PPI and outlined changes to their PPI information requirements for funding bids [45].

In our study, work commitments and location of the group meetings were highlighted as barriers to inclusion of a wider range of members. Due to the COVID-19 pandemic all PPI activity at our Research Unit moved to remote delivery of group work using video conferencing and speaking to individuals on the telephone. This offers an opportunity to try new ways of enabling patient involvement and offer greater flexibility in timing, format, and geographical location of group members. However, this mode of delivery has brought new challenges as it is important to support group members to feel confident using the technology and to consider whether they have access. For individuals with limited internet or device access this may create further barriers to involvement. Further evaluation of these methods will be needed.

Previous literature on the value of PPI has highlighted its role in increasing transparency and accountability of health care research and policy decision-making [19]. This is vital in the wider healthcare context, and ensures that healthcare decisions and treatments are underpinned by a clear evidence base [46]. Transparency has become a key focus in relation to some healthcare innovations that have caused public concern [47]. For instance, transparency in relation to the implementation of new technologies in surgery has been highlighted in recent high profile legal cases such as vaginal mesh treatment and metal-on-metal hip replacements [48, 49]. In our study, value clusters on transparency lacked data within all three systems. This is surprising given increasing public awareness of the issues surrounding transparency in healthcare. A possible explanation is that transparency was viewed as beyond the scope of the PEP-R groups, which is reflected in discussion related to shared decision making and wider impact within the substantive value system. Arguably, transparency is within the responsibility of researchers and healthcare decision-makers. Communicating effectively and clearly with members of the public about how and why healthcare policy decisions are made, and ensuring high quality PPI, is central to processes of transparency. However, these clusters were also not evident within researchers' views. This finding suggests that training and development of PPI activities could focus on the value and delivery of transparency.

## Recommendations for future PPI

This study has highlighted areas for further development of PPI, both in terms of training opportunities and the constitution of group membership. Key areas may include:

- Facilitating time and providing signposting for researchers to engage in training and continued learning about the value and importance of PPI.

- Supporting researchers and members of the group to access information and engage in reflection on the role of PPI in ensuring transparency in research.

- Work to address dimensions of diversity, while also acknowledging that diversity characteristics or experiences are not always be visible.

## Strengths and limitations

### Strengths.

- This research was designed in collaboration with patients. Members of the PEP-R group provided input and the study design was improved by the addition of interviews based on their views.

- The work used an established theoretical framework which enabled exploration of values in the PPI activity.

- Research findings reflect views and experiences of patients who were involved in the groups and researchers who had engaged in PPI activities.

### Limitations.

- The study evaluated an established PPI group with whom researchers had ongoing engagement. To help members of the group and researchers feel able to be open in their responses, data collection was carried out by a new staff member who was fairly unknown to the PEP-R group, but who was still based within the Research Unit team at the time. There was a chance

that the new staff member may not have been considered entirely 'neutral' and for this reason we conducted the study with enhanced anonymity in the data collection and reporting processes. However, this meant that it was not possible to analyse data in light of personal characteristics or other factors that might lead to identification,

- The topic guide based on Gradinger's framework was not designed or reviewed by patient members as they were taking part in the study. Some patients said that at times they found the theoretical nature of the questions difficult to answer. To address this the interviews with patient members were carried out to enable the researcher to provide respondents with the chance to discuss their views in more detail.

## Supporting information

**S1 Table. PEP-R interview topic guide.**
(PDF)

## Acknowledgments

We thank all participants as well as everyone who has been involved in PEP-R, including current and previous patient members of the group as well as colleagues.

## Author Contributions

**Conceptualization:** A. Burston, V. Wylde, R. Gooberman-Hill.

**Data curation:** E. Bradshaw.

**Formal analysis:** E. Bradshaw, K. Whale, A. Burston, R. Gooberman-Hill.

**Investigation:** E. Bradshaw.

**Methodology:** E. Bradshaw.

**Supervision:** R. Gooberman-Hill.

**Validation:** K. Whale.

**Writing – original draft:** E. Bradshaw, K. Whale.

**Writing – review & editing:** K. Whale, A. Burston, V. Wylde, R. Gooberman-Hill.

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
