## [Decision Letter · Decision Letter 0]

5 Jul 2021

PONE-D-21-16524

Value, transparency, and inclusion: Patient involvement in musculoskeletal research

PLOS ONE

Dear Dr. Whale,

Thank you for submitting your manuscript to PLOS ONE. After careful consideration, we feel that it has merit but does not fully meet PLOS ONE’s publication criteria as it currently stands. Therefore, we invite you to submit a revised version of the manuscript that addresses the points raised during the review process.

The comments of the reviewers are overall positive. Yet, they advice for some revisions that I kindly ask you to consider.

We look forward to receiving your revised manuscript.

Kind regards,

Sara Rubinelli

Academic Editor

PLOS ONE

2. Please modify the title to ensure that it is meeting PLOS’ guidelines (https://journals.plos.org/plosone/s/submission-guidelines#loc-title). In particular, the title should be "specific, descriptive, concise, and comprehensible to readers outside the field" and in this case we feel it is not informative and specific about your study's scope and methodology.

Reviewers' comments:

Reviewer's Responses to Questions

**Comments to the Author**

1. Is the manuscript technically sound, and do the data support the conclusions?

Reviewer #1: Yes

Reviewer #2: Yes

2. Has the statistical analysis been performed appropriately and rigorously? 

Reviewer #1: N/A

Reviewer #2: N/A

3. Have the authors made all data underlying the findings in their manuscript fully available?

Reviewer #1: No

Reviewer #2: Yes

4. Is the manuscript presented in an intelligible fashion and written in standard English?

Reviewer #1: Yes

Reviewer #2: Yes

5. Review Comments to the Author

Reviewer #1: Value, transparency, and inclusion: Patient involvement in musculoskeletal research

This is a nice study exploring the views of researchers and patients who are involved in a 'partnership in research' group. It is original research using appropriate methods guided by a relevant theoretical framework. The standard of writing is generally good.

I think the manuscript could be improved with some relatively minor modifications as follows:

Lines 77-79

“However, it is hard to evidence the full scope of PPI impact because the wide range of activities and levels of patient involvement used in research can make it hard to quantify[10, 11].” – Does evidence of impact have to be quantifiable? Perhaps rephrase? Also, avoid using the same terms twice in a sentence (‘hard’).

Lines 86-87

“... researchers have a moral obligation to redress any power imbalance through PPI [15-17].” What power imbalance? In the development of services? In the research or consultation process? Please clarify.

Line 94

“patient involvement work” ?

Lines 119-122

Missing words in this sentence? “We wanted to use a theoretical framework to develop our understanding of PPI once the PEP-R activities were well embedded into the Research Unit and the two specialised groups had been running for several years, [so we???] we carried out a study of the three PPI groups at the Unit 2019 - 2020.”

Line 168

“clinical academics who had attended PPI meetings during the previous three years were” – do you mean PEP-R meetings? In fact, you refer to both PPI and PEP-R groups from here on. It seems you are using them interchangeably? I found that confusing. It seemed to me that you are evaluating your PEP-R group which is an example of PPI.

It would be good if you could provide your questionnaire and interview guide as additional files. This would help readers make sense of why there were clusters with missing data in your framework. Eg. why did you have nothing on accountability/transparency? As you point out, it is surprising.

Lines 189-195

It’s great that you describe this framework but it made for confusing reading. Possibly because the punctuation wasn’t consistent? I found it hard to identify the main construct and the subconstructs. Perhaps you could number the three main constructs? Table 2 shows them clearly – but that comes later.

I think it might be more descriptively accessible to refer to ‘researchers’ rather than ‘staff’ when you ascribe quotes.

Line 224 -223 and lines 289-299

I think these two sections are dealing with the same issue. Can you find a way to discuss them in only one place?

Page 15 – what do staff think about the role of PPI groups in ensuring effectiveness? I find myself wondering if you asked about this. Another reason to supply your questionnaire.

Lines 345-346 – missing words in this sentence?

Line 419 “different degrees of knowledge” knowledge of what? Clinical issues? Research?

Line 426 – “ members did not think that it as their role to make decisions but rather to provide” AS should be WAS

Line 428 - apostrophe - researcher’s should be researchers’

463-476 – can you blend the two sections here? Maybe use one intro and only one of the quotes as it feels repetitious.

548-549 “... or who do not want to be part of a group” How is this changed by online platforms? Can you please clarify.

It’s great that you engage in a discussion about diversity but I’d like to see a more critical engagement with the concept of representativeness / objectivity / generalisability in PPI, perhaps drawing more broadly on the literature on community consultation. Is generalisability (or objectivity) possible? Certainly not in the statistical sense. So what forms of representation are sought, and with what aims?

Thanks for the opportunity to review this manuscript and good luck with the revisions. I look forward to sharing the final paper with colleagues who are setting up a consumer advisory registry in musculoskeletal research.

Reviewer #2: This is a well written manuscript, presented in an intelligible way. I believe that this work is important for and provides good advice for researchers who will be using PPI groups. My only observation would be regarding the data. If it is anonymised then it would be nice if it was more freely available, perhaps included as supplementary information.

6. PLOS authors have the option to publish the peer review history of their article (what does this mean?). If published, this will include your full peer review and any attached files.

Reviewer #1: No

Reviewer #2: No

---

## [Author Response · Author response to Decision Letter 0]

21 Jul 2021

Dear Reviewers, 

Thank you for taking time to review our manuscript and provide constructive feedback. We appreciate the time you have taken and we have provided a table detailing action taken for each point raised. We also include an additional file providing the interview topic guide. 

We hope these changes will meet with your approval. 

Best wishes,

Katie Whale

---

## [Decision Letter · Decision Letter 1]

26 Aug 2021

PONE-D-21-16524R1

Value, transparency, and inclusion: A values-based study of patient involvement in musculoskeletal research

PLOS ONE

Dear Dr. Whale,

Thank you for submitting your manuscript to PLOS ONE. After careful consideration, we feel that it has merit but does not fully meet PLOS ONE’s publication criteria as it currently stands. Therefore, we invite you to submit a revised version of the manuscript that addresses the points raised during the review process.

Reviewers are overall satisfied with the revisions. Yet, there are still some minor aspects to consider before deciding for publication. 

We look forward to receiving your revised manuscript.

Kind regards,

Sara Rubinelli

Academic Editor

PLOS ONE

Journal Requirements:

Reviewers' comments:

Reviewer's Responses to Questions

**Comments to the Author**

1. If the authors have adequately addressed your comments raised in a previous round of review and you feel that this manuscript is now acceptable for publication, you may indicate that here to bypass the “Comments to the Author” section, enter your conflict of interest statement in the “Confidential to Editor” section, and submit your "Accept" recommendation.

Reviewer #1: (No Response)

Reviewer #2: (No Response)

2. Is the manuscript technically sound, and do the data support the conclusions?

Reviewer #1: Yes

Reviewer #2: Yes

3. Has the statistical analysis been performed appropriately and rigorously? 

Reviewer #1: N/A

Reviewer #2: N/A

4. Have the authors made all data underlying the findings in their manuscript fully available?

Reviewer #1: No

Reviewer #2: Yes

5. Is the manuscript presented in an intelligible fashion and written in standard English?

Reviewer #1: Yes

Reviewer #2: Yes

6. Review Comments to the Author

Reviewer #1: (No Response)

Reviewer #2: The revised manuscript has been improved and all comments have now been addressed. There are still one or two grammatical errors that need addressing, but these are very minor.

Lines 96-98

“In assessing the impact of PPI, it has been suggested that understanding theoretical aspects and practical application of patient involvement work provides a fuller picture of how involvement functions”

I’m not sure what this sentence means, please can you clarify.

Lines 99-100

"...one way to achieve this, enabling scope to review the value of PPI and to identify any areas for further development”

Are there words missing from this sentence as it doesn't really make sense?

Lines 101-102

“...identified the existence of a ‘plethora’ of frameworks, tools, and guidelines [? that] exist to support, evaluate, or report PPI”

Are there words missing from this sentence?

This is an interesting, and important piece of work and I will look forward to seeing it published.

7. PLOS authors have the option to publish the peer review history of their article (what does this mean?). If published, this will include your full peer review and any attached files.

Reviewer #1: No

Reviewer #2: No

---

## [Author Response · Author response to Decision Letter 1]

8 Sep 2021

Dear Reviewers, 

Thank you for reviewing our revised manuscript. We have addressed the minor grammatical points you have raised and submit a revised version for approval. Full changes are detailed in the response to reviewers table. 

Best wishes,

Katie

---

## [Editor Report · Decision Letter 2]

15 Nov 2021

Value, transparency, and inclusion: A values-based study of patient involvement in musculoskeletal research

PONE-D-21-16524R2

Dear Dr. Whale,

We’re pleased to inform you that your manuscript has been judged scientifically suitable for publication and will be formally accepted for publication once it meets all outstanding technical requirements.

Kind regards,

Sara Rubinelli

Academic Editor

PLOS ONE
---

## [Editor Report · Acceptance letter]

19 Nov 2021

PONE-D-21-16524R2 

Value, transparency, and inclusion: A values-based study of patient involvement in musculoskeletal research 

Dear Dr. Whale:

I'm pleased to inform you that your manuscript has been deemed suitable for publication in PLOS ONE. Congratulations! Your manuscript is now with our production department. 

Kind regards, 

on behalf of

Dr. Sara Rubinelli 

Academic Editor

PLOS ONE